# Clinical and Radiographic Evaluation of Simultaneous Alveolar Ridge Augmentation by Means of Preformed Titanium Meshes at Dehiscence-Type Peri-Implant Defects: A Prospective Pilot Study

**DOI:** 10.3390/ma13102389

**Published:** 2020-05-22

**Authors:** Carlo Maiorana, Mattia Manfredini, Mario Beretta, Fabrizio Signorino, Andrea Bovio, Pier Paolo Poli

**Affiliations:** Implant Center for Edentulism and Jawbone Atrophies, Maxillofacial Surgery and Odontostomatology Unit, Fondazione IRCCS Ca’ Granda Ospedale Maggiore Policlinico, University of Milan, Via della Commenda 10, 20122 Milan, Italy; carlo.maiorana@unimi.it (C.M.); dr.marioberetta@gmail.com (M.B.); fabrizio.signorino@unimi.it (F.S.); andrea.bovio1@studenti.unimi.it (A.B.); pierpaolo_poli@fastwebnet.it (P.P.P.)

**Keywords:** dental implants, guided bone regeneration, preformed titanium mesh, buccal bone resorption, horizontal bone regeneration

## Abstract

Background: bone augmentation by means of manually shaped titanium mesh is an established procedure to regenerate atrophic alveolar ridges and recreate a proper contour of the peri-implant bone anatomy. Conversely, current literature on the use of preformed titanium meshes instead of traditional grids remains lacking. Therefore, the aim of the present prospective study was to evaluate the use of preformed titanium mesh to support bone regeneration simultaneously to implant placement at dehiscence-type defects from clinical, radiological, and patient-related outcomes. Methods: 8 implants showing buccal dehiscence defects were treated with preformed titanium mesh directly fixed to flat abutments screwed to the implant. Intrasurgical clinical measurements and radiographic evaluations by means of cone-beam computed tomography scans were performed to assess the horizontal bone gain after 8 months from the augmentation surgery. Biological and patient-centered outcomes were also evaluated.; Results: clinically, a mean horizontal bone gain of 4.95 ± 0.96 mm, and a mean horizontal thickness of the buccal plate of 3.25 ± 0.46 mm were found. A mean horizontal bone gain of 5.06 ± 0.88 mm associated with a mean horizontal thickness of the buccal plate of 3.45 ± 0.68 mm were observed radiographically. From a macroscopic aspect, the remodeled graft appeared well integrated with the host bone. Well vascularized newly formed bone-like tissue was observed in intimate contact with the implants. Conclusions: preformed titanium mesh may be effective in supporting simultaneous horizontal bone regeneration at dehiscence-type peri-implant defects. Titanium mesh exposure still remain an issue in this type of surgery.

## 1. Introduction

Dental implantology is a well-established therapeutic alternative to replace missing teeth in fully and partially edentulous patients, providing good survival rates in both pristine and augmented bone in the long term [1,2,3]. Several anatomical and technique-related variables are involved in the maintenance of successful biological, functional, and esthetic outcomes. Notable among these is the horizontal ridge dimension, with particular emphasis on the peri-implant buccal bone thickness (BBT), which strongly impacts the hard and soft tissue stability. In this regard, it has been claimed recently that BBT >1.5 mm should be sought to minimize adverse horizontal and vertical alveolar bone remodeling and provide a protective mechanism against the progression of peri-implantitis [4]. Such experimental findings corroborate available data from previous clinical investigations on the critical role of the BBT. The same BBT threshold has been identified as critical in preserving buccal gingival marginal stability, with more recession in the case of BBT <1.5 mm [5]. Similarly, Marconcini et al. reported more stable peri-implant bone levels and less recessions at the facial soft tissues in case of a thick buccal bone [6]. This complies favorably with Miyamoto et al. who found that BBT >1.2 mm assessed radiographically was associated with significantly lower gingival recessions [7]. Spray et al. observed that patients with BBT <1 to 1.4 mm exhibited the largest horizontal bone loss, while bone resorption decreased significantly as the BBT approached 1.8 to 2 mm, and some evidence of bone gain was seen [8]. Accordingly, Nohra et al. reported a positive correlation between BBT ≥2 mm and minimal marginal bone remodeling in the short term [9]. Taken together, the current evidence clearly indicates that an adequate BBT is of paramount importance when it comes to peri-implant marginal tissue stability. In this respect, guided bone regeneration (GBR) has proven to be a reliable treatment option to obtain favorable BBT around dental implants [10]. If the biological principles of GBR are fulfilled, safe and predictable long-term outcomes have been achieved even in the case of simultaneous implant insertion [11]. As a matter of fact, findings from a systematic review confirmed that immediate implant placement with bone grafting is able to preserve soft tissue stability, horizontal ridge dimension, and BBT [12]. One of the key elements during a bone augmentation procedures is the stability of the blood clot that colonizes the bone graft. This aspect is pivotal since graft incorporation and integration through intramembranous ossification requires stable conditions to occur, avoiding the risk of fibrointegration. To fulfill this task and provide mechanical stability to the bone graft, the use of barrier membranes has been advocated. In this matter, titanium meshes have been extensively used throughout the years to support horizontal and vertical bone formation [13]. Indeed, titanium meshes encompass most of the desirable features a membrane should possess, namely biocompatibility, space-making capability, and clinical manageability [14]. Despite several advantages, one of the main drawbacks is the need to shape and trim the titanium mesh intrasurgically according to the configuration of the defect. This may be a concern in anatomical regions with low accessibility and visibility, resulting in reduced accuracy, difficulties in fixing the device, prolonged surgical time, and increased patient morbidity. To overcome such disadvantages, in recent years the use of preformed titanium mesh screwed directly to the fixture during implant placement and simultaneous localized bone augmentation has been developed [15,16]. The rationale was to combine the advantages of traditional titanium meshes, with the possibility to choose the ideal device conformation according to the defect configuration and fix the titanium mesh directly to the implant with no need for additional stabilization. Furthermore, since pre-shaped titanium meshes are manufactured with round and blunt edges, the risk of membrane exposure due to prominent and sharp edges that may result from intrasurgical manipulations is reduced. However, to the best of the authors’ knowledge, the current evidence regarding the effectiveness of preformed titanium meshes in bone augmentation of localized defects simultaneously with implant insertion is extremely lacking. In view of the aforesaid, the purpose of the present study was to evaluate clinically and radiographically the outcome of horizontal bone regeneration and BBT at peri-implant dehiscence-type defects by means of preformed titanium meshes at single or adjacent implants.

## 2. Materials and Methods

### 2.1. Study Design

This research was designed as a monocentric prospective pilot study performed in a university setting. Patients were consecutively enrolled according to defined criteria and underwent the same research protocol as described below. All surgical and prosthetic procedures were performed by the same clinicians. The patients were informed in detail about possible risks and benefits of the treatment and signed an informed consent before enrollment into the study. The research was conducted in compliance with Good General Practice standards and the current ethical principles for medical research involving human subjects outlined in the Declaration of Helsinki as amended in 2013. The research protocol has been submitted to and approved by the local Ethics Committee (No. 458_2017). The fixtures used in the present study were tapered implants (AnyRidge, MegaGen, Gyeongbuk, Korea) characterized by self-cutting threads and a nanostructured calcium-incorporated surface. The implants featured a 5-mm deep conical connection (10°) combined with an internal hexagon and integrated switching platform. The pre-shaped titanium membranes (i–Gen, MegaGen, Gyeongbuk, Korea) were available in 9 different configurations (type A for incisors/cuspids, type B for premolars, and type C for molars) with different sizes and shapes (small, regular, or wide) and incorporated up to a 100° bend to provide adequate space for bone augmentation. Each titanium mesh had to be fixed to the implant on specially designed flat abutments (i–Gen screws, MegaGen, Gyeongbuk, Korea) of variable height (1–3 mm), by means of a cover screw. The timepoints of the protocol were as follows: presurgical evaluation (T0), surgical stage with implant positioning and simultaneous bone augmentation procedure (T1), and re-entry stage after 8 months from implant placement and simultaneous bone regeneration (T2).

### 2.2. Patient Selection

Between October 2017 and March 2018, a total of 5 consecutive patients were enrolled. The patients were referred to the authors’ department seeking implant rehabilitation of partially edentulous ridges in the posterior mandible or maxilla. Patients were identified as candidates for dental implant treatment based on clinical and radiographic examinations (Figure 1).

The main inclusion criteria included single or multiple maxillary or mandibular partial edentulism in the premolar and molar sectors; the need for implant rehabilitation; tooth loss/extraction occurred ≥2 months; age ≥18 years at the time of surgery. Additionally, patients were included if presenting inadequate width of the residual alveolar process, with the need for horizontal bone augmentation of at least 3 mm assessed in a cone-beam computed-tomography (CBCT) scan in order to obtain ≥1 mm of bone circumferentially around the implant at the end of the treatment. Further inclusion criteria such as, the possibility to insert the fixture in a prosthetically guided position simultaneously with bone augmentation, primary stability of the implant, and vertical soft tissue thickness ≤2 mm according to Linkevicious et al. [17], were evaluated intra-surgically. In the event that the patient was found ineligible at the time of surgery, the subject was excluded from the study and a two-stage procedure was carried out following good clinical standards. The exclusion criteria included the need for a two-stage procedure, full-mouth plaque and bleeding scores ≥25%; abuse of alcohol or drugs; presence of acute oral infections; remote or recent radiation therapy in the oromaxillofacial area; recent chemotherapy; uncontrolled diabetes; pregnancy; smoking habit (>10 cigarettes/day); and any other drug therapy, systemic disease or syndrome that may affect hard and soft tissue biology. In addition, patients with a lack of collected data at any stage of the planned follow-up were also excluded.

### 2.3. Pre-Surgical Phases

Following patient selection, a CBCT scan (CS 9300, Carestream, Atlanta, GA, USA) was performed before surgery (T0) to precisely assess the anatomy of the residual alveolar process and select the most appropriate implant dimensions and titanium mesh configuration. A preliminary wax-up was performed on stone casts in order to realize the surgical stent according to the analogic prosthetic project. All patients received a professional oral hygiene session one week prior to surgery. In addition, oral hygiene instructions were reinforced, and chlorhexidine 0.2% mouth rinses twice a day for one week were prescribed.

### 2.4. Surgical Phases

The surgical procedure (T1) was performed by the same surgeon on an outpatient basis. Antibiotic therapy consisting of amoxicillin clavulanate acid (GSK, Brentford, UK) 2 g/day for 6 days, or clindamycin (Abbott, Chicago, IL, USA) 500 mg/day for 7 days in case of penicillin-allergic patients, was prescribed to reduce the risk of infections, starting with 2 g one hour before surgery. Anti-inflammatory therapy with Ketoprofen Lysine (Dompè, Milan, Italy) 80 mg/day for 3 days, starting with 80 mg 1 h before surgery, was also recommended. Local anesthesia was induced with mepivacaine 2% (Molteni Dental, Milan, Italy) with epinephrine 1:100.000. An intramuscular injection of 8 mg of dexamethasone phosphate (LFM, Milan, Italy) was performed to reduce the postoperative swelling. A full-thickness buccal flap was raised following mid-crestal incision in the maxilla, whereas in the mandible the incision was carried out taking care to divide the keratinized gingiva into equal parts on the buccal and lingual sides (Figure 2). Vertical releasing incisions were performed if necessary.

The vertical thickness of the soft tissues was measured using a 1-mm marked periodontal probe (PCP-UNC 15, Hu-Friedy, Milan, Italy). If the vertical soft tissue thickness was ≤2 mm, the tissue was considered thin and the patient was excluded from the present study.

At this point, the osteotomy site was prepared at 800 rpm with copious irrigation with sterile saline according to the manufacturer’s instructions in a prosthetically guided position. When adjacent implants were positioned, direction pins were used to ensure parallel insertion of the implants, maintaining an inter-implant distance of 3 mm. All the fixtures were inserted using a torque-controlled driver at 60 rpm and finally positioned 0.5 mm subcrestal to the crestal edge (Figure 3). A torque value between 30 and 50 Nm was deemed acceptable, depending on the bone density.

Subsequently, a flat abutment of variable height (1–3 mm) was connected to the fixture depending on the ridge morphology: a standard 1-mm cuff height was used in case of sufficient vertical space, but 2 mm or 3 mm cuff heights were also chosen according to the clinical situation. The graft consisted in autogenous bone chips harvested nearby the recipient site with a manual bone scraper (Safe scraper, META, Reggio Emilia, Italy) and deproteinized bovine bone mineral (DBBM) particles (Bio-Oss, Geistlich Pharma AG, Wolhusen, Switzerland) in a ratio of approximately 1:1. Then, the appropriate titanium mesh was selected according to the shape and size of the defect, and fixed to the implant with the cover screw (Figure 4).

Each titanium membrane was prepared in order to fit the individual anatomy and to contain the graft. The autogenous bone chips were placed in intimate contact with the exposed surface of the implant, whereas the DBBM particles were grafted to fill the remaining space between the defect and the titanium mesh. Periosteal releasing incisions were performed to passivate the flaps, and a tension-free first intention healing was accomplished with horizontal mattresses and interrupted non-resorbable sutures (CV-5 and CV-7, Gore-Tex^®^; W.L. Gore and Associates, Flagstaff, AZ, USA). An intra-oral radiograph was finally performed and ice-packs were provided. After surgery, all the patients observed a 10-day liquid diet, followed by an additional 15 days of soft foods. Immediate post-surgical oral hygiene protocol included 0.2% chlorhexidine rinsing three times a day up to one week following suture removal. Postoperative pain was controlled by administering Ketoprofen Lysine 80 mg/day. Sutures were removed after 14 days. All the patients were advised against using removable partial dentures with mucosal support over the surgery site. Eight months later, the re-entry surgery was performed to assess the results of the reconstructive procedure.

### 2.5. Re-Entry Surgery and Prosthetic Phases

The re-entry surgery (T2) was scheduled 8 months after the implant placement and bone augmentation procedure according to Tallarico et al. [18]. In brief, a mid-crestal incision was made to elevate a mucoperiosteal flap and expose the augmented ridge (Figure 5). The cover screw together with the titanium mesh and the flat abutment were removed and a transmucosal healing abutment of adequate width and height was connected to the implant. Finally, the flaps were adjusted to fit around the neck of the healing abutment and sutured. In this phase, if necessary, soft tissue augmentation by means of deepithelialized connective tissue graft was performed in order to improve the buccal soft tissue thickness and optimize the aesthetic. Sutures were removed after 2 weeks. A post-operative CBCT scan was performed.

After additional 2 weeks, alginate impressions were taken to create individualized impression trays, which were then used to take secondary impressions using a polyether material and pick-up technique. One week later, screw-retained temporary resin restorations were applied (Figure 6).

The provisional rehabilitations were left for a period of 3 months, after which the definitive screw-retained metal-ceramic restorations were delivered (Figure 7).

Occlusion was checked to remove any pre-contacts or interferences in centric, lateral, or protrusive movements. Professional oral hygiene maintenance care was provided every 4 months the first year, and then twice a year thereafter.

### 2.6. Study Outcomes

The variables of the study were strictly related to the evaluation of the bone augmentation procedure. The main outcome was the qualitative and quantitative analysis of the regenerated buccal bone from both clinical and radiological perspectives. The quantitative measurements were collected by the same clinician under identical conditions. The secondary outcome was the clinical assessment of early and late biological complications at the augmented site.

### 2.7. Clinical Assessment

The clinical measurements were performed at T1 and T2 with a 1-mm marked periodontal probe (PCP-UNC 15, Hu-Friedy, Milan, Italy) and a caliper. The width of the residual crest (mm) was registered before implant placement and at the re-entry surgery in correspondence of the mesio-distal center of the ridge with respect to the adjacent teeth. The defect configuration was assessed after implant placement at T1. The vertical defect height (mm) was measured from the implant shoulder to the first bone-to-implant contact (BIC); the horizontal defect width (mm) was measured from the mesial to the distal bone crests. The same measurements were registered at T2 in case of a residual defect. At T2, the buccal bone thickness (mm) was measured from the implant neck to the vestibular wall in a direction perpendicular to the long axis of the implant. All the measurements were rounded to the nearest half millimeter. The qualitative clinical evaluation of the regenerated bone was performed at the re-entry surgery following removal of the titanium mesh. The following parameters were considered: integration, stability, and vascularization of the graft.

### 2.8. Radiographic Assessment

The radiographic linear measurements were performed in the same sagittal section at T0 in correspondence of the future implant location, and at T2 (Figure 8).

In order to standardize the measurement position in both CBCT scans, the orthoradial cross-sectional slice showing the entire implant diameter was selected as the initial reference image. After performing the linear measurements at T2, the mesio-distal distance between this slice and the first-visible image showing the adjacent mesial or distal natural tooth was registered and reported in the CBCT scan performed at T0. At this point, in the pre-operative CBCT scan, it was possible to select the corresponding slice at the same mesio-distal distance from the reference tooth. This allowed performing the linear measurements at T0 approximately in the same location as that selected at T2.

The following variables were registered according to the clinical measurements: residual crest width (mm) at T0; augmented crest width (mm) at T2; buccal bone thickness (mm) at T2. All measurements were calculated 1 mm apically to the bone crest, where the implant shoulder was placed as recommended by the manufacturer’s recommendations. The digital image of each sagittal section was imported with a resolution of 1200 dpi in a dedicated software (ImageJ 1.49v, Research Services Branch, National Institute of Health, Bethesda, MD, USA). The calibration of the pixel/millimeter ratio was performed on the basis of a known distance, namely the calibrated pixel/millimeter ruler reported in each CBCT sagittal section.

### 2.9. Biological Complications

To evaluate post-operative drawbacks, a distinction was made between early and late biological complications. Early complications were those that occurred within 2 weeks from the surgical procedures, and included discomfort/pain, edema/swelling, and extraoral hematoma. From 2 weeks to T2, late biological complications included infection, titanium mesh exposure, partial or complete loss of the graft, and implant mobility at the re-entry surgery.

### 2.10. Statistical Analysis

Data were entered into tables and evaluated using a statistical software (IBM SPSS Statistics version 24, Armonk, NY, USA). Patient demographics and the distribution of implants were analyzed using descriptive statistics. Means and standard deviations were calculated for quantitative variables, such as patient age, initial horizontal width of the alveolar crest, horizontal gain of the alveolar ridge width, and thickness of the buccal bone. Percentages of biological complications were calculated for qualitative categorical variables. The surgical site was adopted as the statistically independent unit.

## 3. Results

Overall, 8 surgical sites distributed in 5 patients (2 females and 3 males, mean age at the time of surgery: 65.4 ± 8.3 years) were available for the analyses. Demographic data of patients, and implant and titanium mesh characteristics at each surgical site were reported in Table 1.

With respect to early biological complications, extraoral hematoma was observed in two patients (25% of surgical sites), edema/swelling was reported by one patient (12.5% of surgical sites), while none of the subjects complained of discomfort/pain. In regards to late biological complications, the healing proceeded uneventfully in the majority of patients. A minor complication was observed in one patient (12.5% of the surgical sites), who reported loss of the cover screw after 3 months from the surgical procedure. At the clinical examination, the soft tissues appeared healthy with no evidence of infection, soft tissue dehiscence, or titanium mesh exposure. The graft and the titanium mesh appeared firm and stable upon palpation. The re-entry procedure was carried out according to the scheduled timing protocol after 8 months from the surgical phase with no further complications. Another patient experienced major complications, namely bilateral exposure of the titanium mesh at the occlusal-lingual aspect of both mandibular sites after 4 months. The complication was managed with gentle cleaning of the area with an extra soft toothbrush soaked in 1% chlorhexidine gel. In addition, the patient was asked to apply 1% chlorhexidine gel, 2 times per day, and was instructed to rinse with 0.12% chlorhexidine, 2–3 times per day. Regular follow-ups were scheduled every 2 weeks up to the re-entry phase. No evidence of pain, graft infection, and mobility of the titanium mesh were found during the remaining healing time although the exposure remained stable but complete resolution was never achieved. Only partial unscrewing of the exposed cover screws was occasionally observed during the follow-up recalls. In such cases, the cover screws were re-tightened with a screwdriver. The overall exposure rate was 25%, however the planned bone regeneration was achieved in all cases. The clinical measurements at each surgical site were represented in Figure 9.

The mean width of the residual alveolar ridge at the coronal portion of the crest measured at T1 was 3.43 ± 0.77. After implant insertion, the analysis of the defect configuration showed a mean vertical defect height of 3.5 ± 0.75, and a mean horizontal defect width of 3.5 ± 0.50 mm. At T2, the mean crestal width of the augmented alveolar ridge was 8.38 ± 0.58 mm, corresponding to a mean horizontal bone gain of 4.95 ± 0.96 mm. The mean horizontal thickness of the buccal plate was 3.25 ± 0.46 mm. In most of the patients, no residual defects were found at T2. Only the patient experiencing titanium mesh exposure showed a residual dehiscence of approximately 1 mm in height and 0.5 mm in width at the buccal aspect of both surgical sites. The radiographic measurements at each surgical site were represented in Figure 10.

The mean width of the residual alveolar ridge 1 mm apically to the crest measured at T0 was 3.45 ± 0.68 mm. At T2, the mean width of the augmented alveolar ridge measured 1 mm apically to the crest was 8.52 ± 0.56 mm, corresponding to a mean horizontal bone gain of 5.06 ± 0.88 mm. The mean horizontal thickness of the buccal plate measured 1 mm apically to the crest was 3.45 ± 0.68 mm. From a qualitative point of view after titanium mesh removal at T1, all implants appeared clinically stable. In all patients the remodeled graft appeared well integrated with the host bone. Bleeding newly formed bone-like tissue was clinically observed after mesh removal. Remnants of DBBM particles incorporated into the regenerated tissue were occasionally observed. The anatomy of the augmented bone closely resembled the shape of the titanium mesh. Periosteum-like tissue interposed between the titanium mesh and the regenerated bone was observed, with no fibrous encapsulation of the biomaterial in the augmented bone. Clinically, this layer appeared as a dense and poorly vascularized connective soft tissue. The thickness of the pseudo-periosteum measured approximately 0.5 to 1 mm in sites healed with no post-operative complications. Conversely, in both sites exhibiting exposure of the titanium mesh, the thickness of the pseudo-periosteum increased up to 1.5 to 2 mm. This finding was compatible with a bone reaction aimed to protect the newly formed tissue and to facilitate secondary-intention healing following membrane exposure. The fact that, in both sites, the occlusal-lingual aspect of the barrier was directly exposed to the oral environment may have led to micromovements that further contributed to the formation of a thicker pseudo-periosteum.

## 4. Discussion

The healing of the alveolar process following tooth loss is characterized by a substantial reduction of the original ridge width, with a greater bone remodeling at the buccal aspect [19]. As a consequence of this socket-healing pattern, prosthetically guided implant insertion may likely results in peri-implant dehiscence-type defects at the expense of the buccal plate. To manage this complication, preformed titanium mesh was used in the present study at each defect to stabilize the bone graft and prevent soft tissue collapse. The aim was to increase the width of the residual ridge and obtain an adequate amount of BBT following implant insertion. The biological rationale was that bony dehiscence defects ≤5 mm are more susceptible to vertical bone loss at the buccal aspect and marginal bone remodeling if left for spontaneous healing compared to grafted defects [20]. A similar behavior could have been expected in the present study, as both width and height of the dehiscence-type defects were constantly ≤5 mm. Bone augmentation by means of autogenous bone and DBBM secured by a preformed titanium mesh yielded an overall horizontal augmentation of approximately 5 mm, with a BBT of more than 3 mm. This increase has been verified clinically at the level of the bone crest, and 1 mm apically to the bone crest in correspondence with the original coronal-apical placement of the implant shoulder. These results are in full agreement with those obtained by Tallarico et al., who measured a mean horizontal bone gain of 5.06 ± 1.13 mm in the case of simultaneous implant placement and bone reconstruction with preformed titanium mesh secured to the implant [18]. The results presented herein are also comparable with those reported by Zita Gomes et al., who found a mean horizontal bone gain of 3.67 ± 0.89 mm following bone augmentation at dehiscence-type defects with the same model of preformed titanium mesh [15]. It is worth noting that, by contrast with the present investigation, the authors grafted the defects with DBBM alone. The question whether the use of DBBM is not inferior to the use of DBBM and autogenous bone chips for the treatment of bony dehiscences at implant placement has been addressed in a recent randomized controlled trial [21]. The results indicated that horizontal defect width changes at the implant shoulder did not differ significantly between the two groups. This strengthened the importance of graft stabilization irrespective of the type of material used according to the comparison between our results and those found by Zita Gomes and coworkers. It is noteworthy that the authors in both groups covered the graft with a resorbable native collagen membrane without fixation devices such as pins or sutures. This may have contributed to a significant contraction of the graft of >50% after 4 months of healing, resulting in suboptimal dehiscence coverage around the implant neck. Such finding merely reinforces the need for stiff and stable barriers covering the graft during the healing period. Indeed, in the present study the augmented buccal bone height was present at the level or, in some cases, even above the top of the implant shoulder. One may speculate that the use of flat abutments allowed stabilizing the titanium mesh at least 1 mm above the implant neck, thus overcorrecting the defect in a vertical dimension, and preventing the collapse of the membrane toward the implant shoulder. A similar outcome has been detected in case of implant placement and simultaneous bone augmentation using L-shaped titanium mesh stabilized with sutures for localized defects [22]. Following implant placement, healing caps or cover screws were connected to support the barrier vertically. Interestingly, 15 out of 16 sites showed an average remaining labial bone height above the top of the implants of 1.13 ± 1.30 mm, which is concordant with the augmented bone anatomy observed in the present research. The surgical protocol contemplated the use of collagen membranes to cover the titanium mesh as an additional barrier for better bone regeneration. This differed from the present study and may constitute a variable, as no resorbable membranes were used over the grids. In this regard, a focused animal study evaluated the effect of collagen membranes laid over preformed titanium meshes connected to the implants in case of implantation and simultaneous GBR [23]. Data from the histological analysis rejected additional benefit for mucosal healing and buccal bone preservation when collagen membranes were used to overlay the titanium meshes. Accordingly, preformed titanium meshes alone increased mechanical properties in terms of higher mechanical stiffness and fatigue resistance compared to manually-shaped titanium grids, and achieved better histological results considering percentages of new bone area, bone-to-implant contact, distance from the new bone to the old bone, and distance from the osseointegration to the old bone compared to collagen membranes [24]. Conversely, platelet-rich fibrin [25] and platelet-rich plasma [26] have been specifically used to cover titanium meshes and improve soft tissue healing. The rationale was to deliver stem cells, fibrin, platelets and leucocytes embedded in such hemoderivatives in order to enhance micro-vascularization and migration of epithelial cells. Interestingly, in both aforementioned studies, the use of platelet-rich fibrin or platelet-rich plasma respectively, provided significantly less exposures of the titanium meshes compared to sites not treated with hemoderivatives. These finding cannot be confirmed in the present study due to a lack of control group in this sense, however the clinical outcome supported the use of titanium mesh alone during bone augmentation procedures. A visual inspection has been conducted in this study to evaluate macroscopically the quality of the augmented bone at the re-entry surgery performed after 8 months from the reconstructive procedure. After elevating the flap, the external layer of the titanium mesh appeared to be surrounded by a dense fibrous tissue with no signs of inflammation. The device appeared to adhere mildly to the newly formed tissue and was, therefore, removed with no difficulties. A whitish soft tissue was present underneath, and the space under the grid was completely filled by hard tissue with the macroscopic features of newly formed bone. The graft appeared well maintained and incorporated into the native bone and in intimate contact with the implant. These clinical findings have been commonly observed after uneventful healing of augmented ridges by means of titanium grids [24]. Actually, such macroscopic features have been corroborated by histological analyses performed after 4 months from bone regeneration with preformed titanium mesh connected to the implant [16]. The authors observed a thin connective tissue layer underneath the titanium mesh, well-incorporated biomaterial particles surrounded by 5% of connective tissue and 80% of newly formed vital bone. Inflammatory cell infiltrates made up the remaining 15%. It is safe to assume that after a healing period of 8 months waited in the present investigation, the macroscopic features observed may reflect the good histological outcomes reported after 4 months. Titanium mesh exposure is one of the main concerns of this technique, with an average complication rate ranging from 16.1% [13] to 34.8% [27]. Although membrane exposure is caused by multiple factors, prominent, sharp, and cutting edges resulting from the contouring of the titanium mesh may play a certain role in the occurrence of such complication. To overcome such drawback, the preformed mesh used in the present study is fabricated with round and blunt edges to prevent mucosal irritation. Nevertheless, in 2 out of 8 surgical sites exposure of the titanium mesh was observed. One may expect a partial loss of the graft, according to previous findings claiming that in case of peri-implant dehiscence defects, the sites with non-resorbable membrane exposure had 27% less defect reduction than the sites without exposure [28]. Accordingly, Lizio et al. found a significant negative correlation between the amount of reconstructed bone and area of titanium mesh exposed accounting for approximately 30% of the overall planned bone volume [29]. In the present study, the exposures occurred at the lingual aspect of the ridge, not in close proximity with the bone graft. Furthermore, the patient was regularly recalled for plaque control and was instructed to use chlorhexidine gel and rinsing solution during the remaining healing time. All of these factors taken together may explain why titanium mesh exposures did not jeopardize the outcome of the bone regeneration in this subject, who showed a BBT of 2.98 mm and 3.31 mm at 8 months. Another very important advantage is that this material tolerates exposure very well. Louis et al. demonstrated that, in spite of exposure of the membrane in 23 (52%) of their patients treated with titanium mesh, only one patient had failure of the graft [30]. The present study has some limitations. First of all, the BBT has been assessed also on CBCT scans. Although this is consistent with the aforementioned studies [15,18,22], such measurements may not be precise. The available evidence suggests that a certain mismatch between the real and the measured BBT adjacent to titanium implants should be expected. The inaccuracy might range between an overestimation of +0.5 mm [31] and an underestimation of -0.3 mm [32]. Nonetheless, taking into account this range of values and the clinical measurements, the BBT obtained in the present study constantly reached a safe dimension of ≥2 mm. The main limitation of the present study is the small number of patients included. In addition, the sample of patients enrolled consisted of a conveniently sampled population that was treated in a university setting under meticulous surgical procedures and professional oral hygiene maintenance regimen. Furthermore, in the present study knife-threaded implants were placed in order to gain adequate primary stability to support the titanium mesh. This type of implant may be more indicated than others in compromised anatomical situations as enough primary stability can be achieved with low torque values [33]. All of these concerns recognize a lack of external validity and demand that the reported results should be interpreted with caution and should not be extrapolated to the general population. This limitation can be solved by performing further trials with larger population, that could be calculated based on the preliminary results reported herein.

## 5. Conclusions

Within the limits of the present study, results suggest that preformed titanium mesh may be considered a valid tool to support bone regeneration at dehiscence-type peri-implant defects. An adequate amount of horizontal bone gain and BBT can be achieved predictably. Patient-related variables indicated that the procedure was well tolerated by all patients. Titanium mesh exposure still remains an issue in this type of surgery.

## Figures and Tables

**Figure 1 materials-13-02389-f001:**
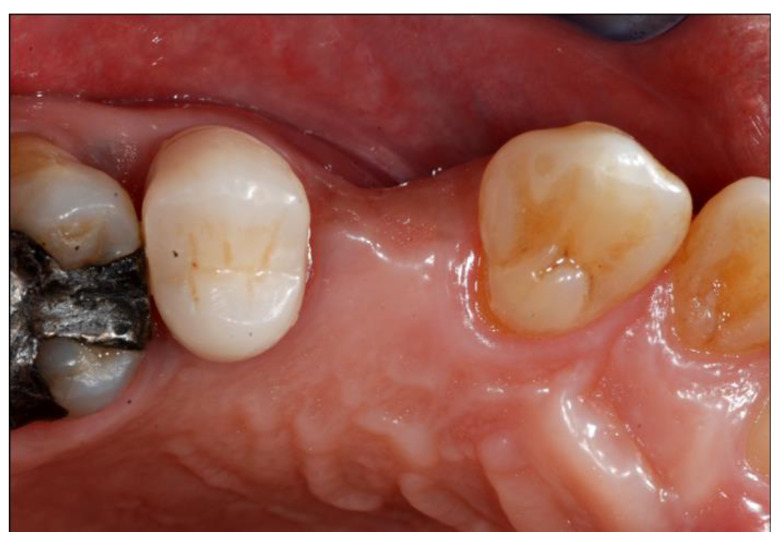
Presurgical clinical situation, occlusal view of the defect.

**Figure 2 materials-13-02389-f002:**
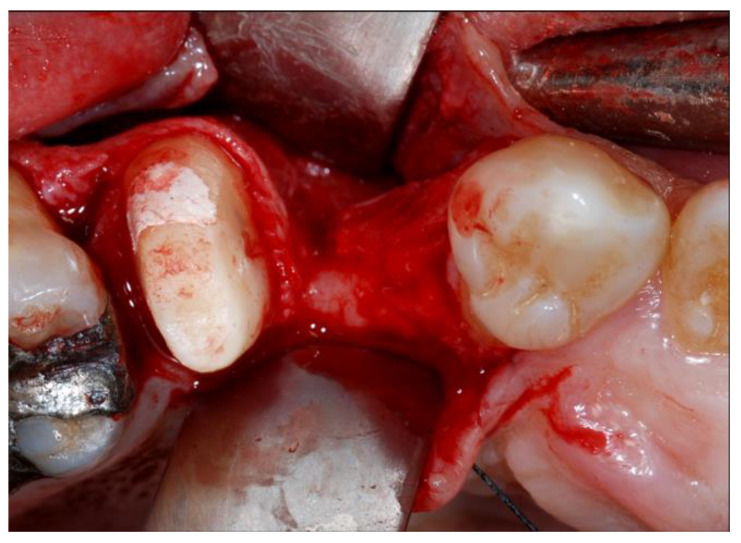
Intraoperative view of the surgical defect after elevation of full-thickness flap.

**Figure 3 materials-13-02389-f003:**
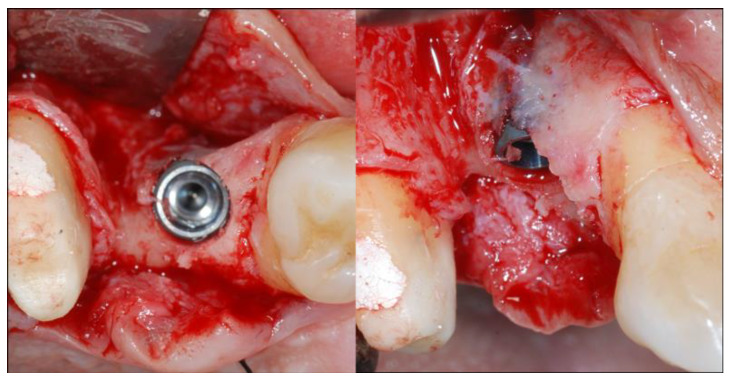
On the left the occlusal view, on the right the vestibular view of the dehiscence of the thin buccal plate after implant insertion.

**Figure 4 materials-13-02389-f004:**
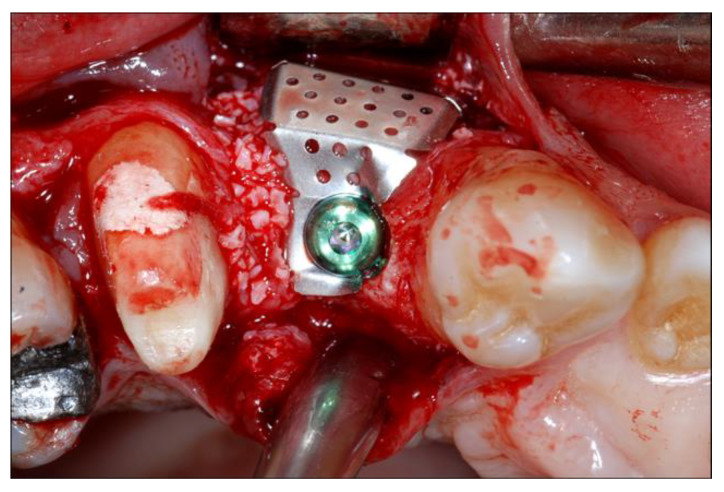
Placement of the titanium mesh, stabilized to the implant with a cover screw fixed to the flat abutment.

**Figure 5 materials-13-02389-f005:**
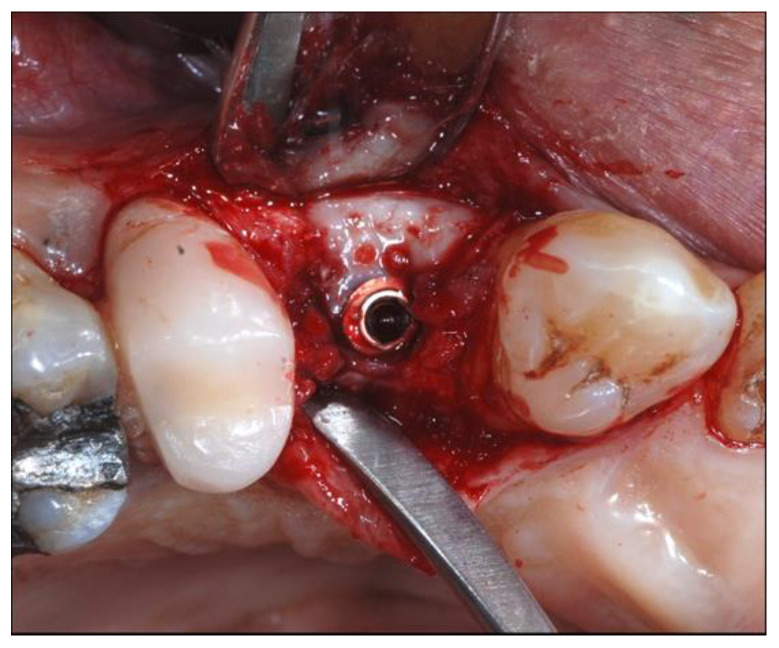
Occlusal view of the bone gain after 8 months from the augmentation surgery at the re-entry phase.

**Figure 6 materials-13-02389-f006:**
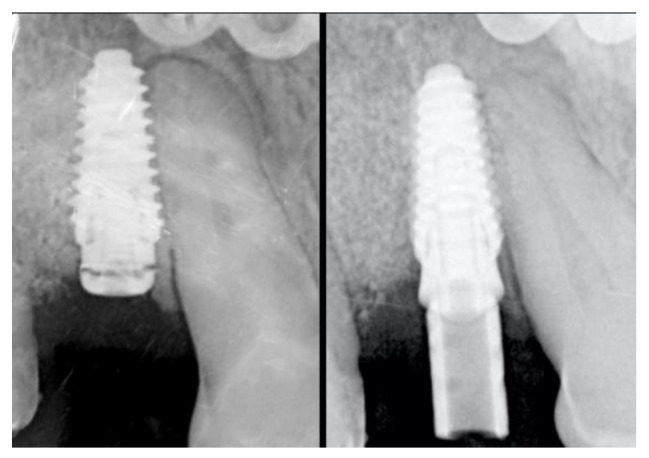
On the left Radiological evaluation at the re-entry surgery, on the right at provisional delivery.

**Figure 7 materials-13-02389-f007:**
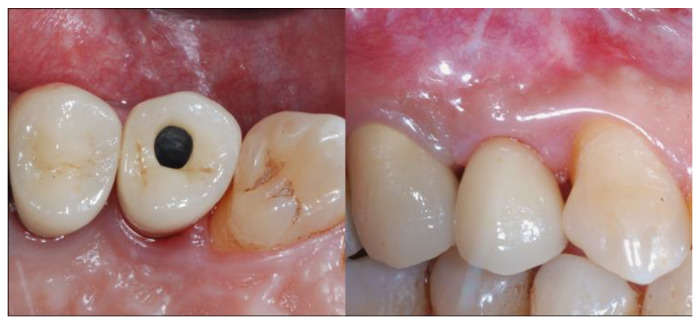
Delivery of the definitive screw-retained metal-ceramic rehabilitation: on the left the occlusal view and on the right the buccal view.

**Figure 8 materials-13-02389-f008:**
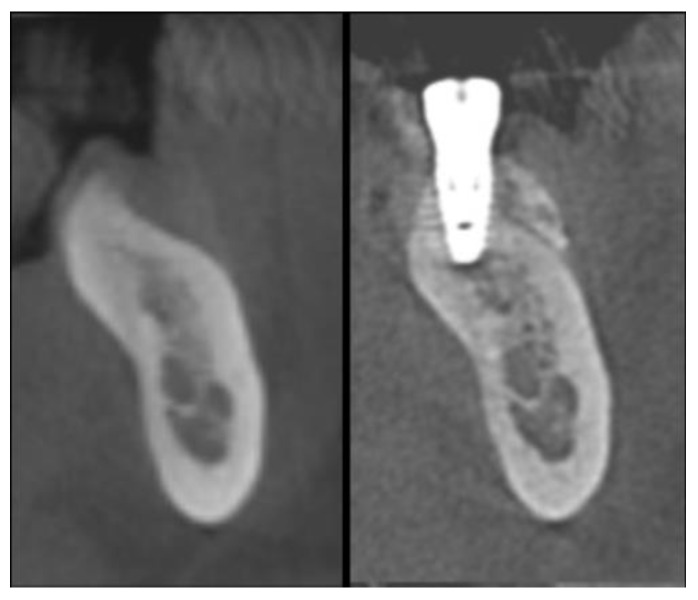
Cone beam computed tomography (CBCT) scans of a mandibular site before surgery (left side) and after removal of the titanium mesh (right side), 8 months after the reconstructive procedure.

**Figure 9 materials-13-02389-f009:**
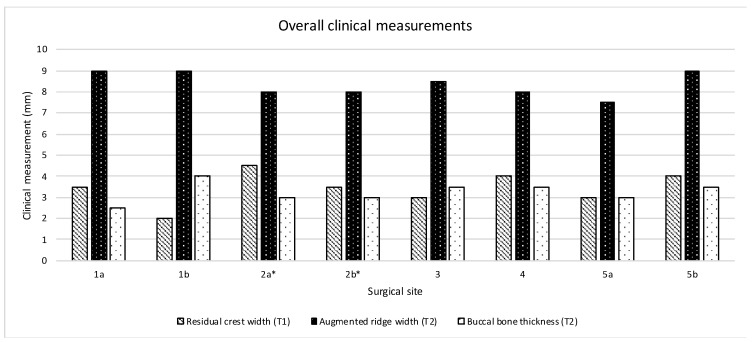
Graphical representation of the clinical measurements registered at each surgical site during the surgical procedure (T1) and after 8 months (T2); *surgical site that showed membrane exposure.

**Figure 10 materials-13-02389-f010:**
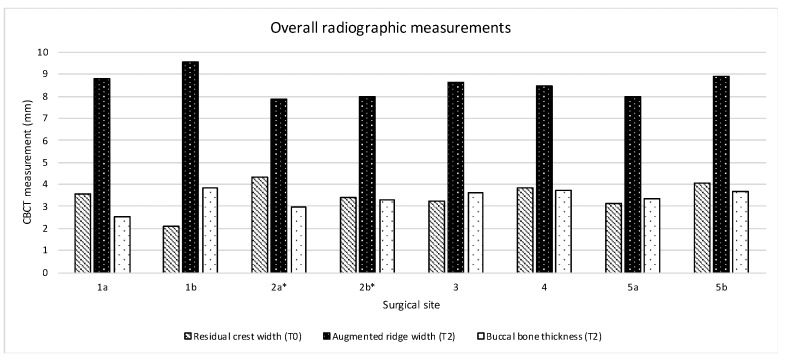
Graphical representation of the radiological measurements calculated at each surgical site before the surgical procedure (T0) and after 8 months (T2); *surgical site that showed membrane exposure.

**Table 1 materials-13-02389-t001:** Demographic distribution and characteristics of patients and implants based on the surgical sites. *Surgical sites that showed titanium mesh exposure.

Patient ID	Site ID	Sex	Age	Surgical Site	Implant Dimension (mm)		Torque (Ncm)	Dehiscence Defect (mm)		Grid Configuration
					Diameter	Length		Height	Width	
1	1a	M	69	34	4	10	50	4	3	A1
	1b			36	4	8.5	50	3	4	B2
2	2a*	F	65	36	4	8.5	35	5	3,5	B1
	2b*			46	4	10	35	3	4	B1
3	3	M	50	14	4	8.5	50	3	4	B1
4	4	M	75	15	3.5	8.5	35	3	3	A1
5	5a	F	68	45	3.5	7	50	4	3	A1
	5b			46	7	7	50	3	3.5	B1

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
