# Peer review of "Clinical and Radiographic Evaluation of Simultaneous Alveolar Ridge Augmentation by Means of Preformed Titanium Meshes at Dehiscence-Type Peri-Implant Defects: A Prospective Pilot Study"

_materials, 2020, doi:10.3390/ma13102389_

Round 1

Reviewer 1 Report

Very interesting topic to clinicians in spite of small sample size. 1) Most reconstructed sites by titanium mesh are reported to be covered by a thin perisoteum-like soft tissue and do you also have any finding about this, such as thickness or change after the mesh exposure? 2) Some surgeons prefer covering the titanium mesh by PRF or a collagen membrane to augment the thickness of overlying soft tissue? If you have any idea about this, please add in your manuscript. 3) If you have any plain dental X-rays, please add them in your manuscript. Only a few CT scans can be hard to understand the total therapy procedure clearly to some clinicians. Thanks.

Reviewer 2 Report

Given that this being a prospective pilot trial, the authors have achieved interesting results with the use of a preformed non-resorbable titanium mesh. The paper is generally well written and supported with clinical images.

I have some queries as outlined below:

1. Figure 4: It looks like all the graft material was not covered by the Titanium mesh. Was any resorbable/other membrane used to cover the exposed graft or the soft tissue was sutured over it directly?

2. Any reasons for choosing the 8 month time-point for re-entry?

3. Although time-points T0,1,2 are mentioned in the respective paragraphs, it would be good mention about these time-points separately at the beginning of the Methods section.

4. Page 7, Lines 208-210 - "Qualiative assessment of regenerated bone". Was the assessment done clinical only? Any histomorphometric or microscopic analysis?

5. For the CBCT measurements, was any superimposition performed between preop and postop CBCTs. In my opinion this would have been an ideal way to measure the bone width across different time-points.

6. Page 8, lines 242-244: Minor complication - Loss of cover screw at 3 months. Wasn't submerged healing utilised in this study protocol? If yes, this would mean that there was a soft tissue dehiscence which might have created a space for the over screw to come out. Was this due to a soft tissue collapse following surgery?

7. Page 9, Lines 274-278: It is mentioned that 'well vascularised newly formed bone like tissue was observed in intimate contact with implants'. In my opinion the only way to assess this is histologically. Please consider modifying the statement.

8. Was soft tissue grafting performed in any of the study patients?
